# Evaluation of the Multiple Tissue Factors in the Cartilage of Primary and Secondary Rhinoplasty in Cleft Lip and Palate Patients

**Dace Buile** [1,*], **Mara Pilmane** [1] and **Ilze Akota** [2]

1 Department of Morphology, Institute of Anatomy and Anthropology, Riga Stradiņš University, 9 Kronvalda Str., LV-1010 Riga, Latvia

2 Department of Maxillofacial Surgery, Institute of Stomatology, Riga Stradiņš University, 20 Dzirciema Str., LV-1007 Riga, Latvia

* Correspondence: dace.danberga@gmail.com; Tel.: +37-126-445-444

**Abstract:** Cleft lip and palate (CLP) is one of the craniofacial defects. The objective of this study was to identify the differences in appearance between the tissue factors in cartilage of CLP patients after primary and secondary rhinoplasty. Immunohistochemistry was performed with MMP-2, MMP-8, MMP-9, TIMP-2, IL-1α, IL-10, bFGF, and TGFβ1. The quantification of the structures was performed using a semi-quantitative census method. MMP-2, -9, IL-1a, and bFGF demonstrated higher number of positive cells in patients, while the number of MMP-8, IL-1a, -10 and TGFβ1 cells was higher or equal in the control subjects. The only statistically significant difference between CLP-operated patients was found in the TIMP-2 group, where the primary CLP patient group had a higher number of TIMP-2 positive chondrocytes than the secondary CLP patient group (U = 53.5; $p$ = 0.021). The median value of the primary CLP group was ++ number of TIMP-2 positive chondrocytes compared to +++ in the secondary CLP group. No statistically significant difference was found between primary and secondary rhinoplasty patients for other tissue factors. Commonly, the rich expression of different tissue factors suggests a stimulation of higher elasticity in cleft affected cartilage. The statistically significant TIMP-2 elevation in primary operated cartilage indicates an impact of the selective tissue remodeling for hard tissue.

**Keywords:** CLP; cartilage; rhinoplasty; immunohistochemistry

## 1. Introduction

The second most common congenital craniofacial deformity is cleft lip and palate (CLP), which is caused by abnormal soft and hard organogenesis during embryonic development. The incidence of cleft lip with versus without cleft palate (CL/P) is 1:1000 [1,2]. The goal of early postnatal surgical procedures is to promote adequate palatal shelf growth and fusion. New opportunities for encouraging tissue growth at the site of the surgical repair for palatal clefts have emerged due to the advances in scaffold-based delivery technologies for precision tissue engineering [3]. Asymmetries often shift as a child grows and develops, making the nasal deformity a daunting challenge. The repair of absent or asymmetric cartilage and the replacement of bone components are crucial for a successful treatment of the cleft lip and palate patients [4,5]. Avascular cartilage is mostly made up of extracellular matrix (ECM), which is maintained by a tiny number of local chondrocytes. The ECM is sustained by an array of growth factors and cytokines in healthy tissue [6]. Despite the fact that multiple tissue factors are thought to have a role in the morphopathogenesis of CLP, research on human cleft nasal septum cartilage is limited due to ethical issues and a lack of available material.

Up to date, there are relatively few studies and data on remodeling factors, resorption factors, growth factors, and cytokines of cartilage due to the difficulty of acquiring hyaline cartilage of the nasal septum because of the ethical considerations.

Matrix metalloproteinases (MMPs) expression and their natural inhibitors (TIMPs) in craniofacial development is tissue specific. MMPs and TIMPs are believed to be necessary for the growth of the mammalian palate [7].

MMPs degrade the ECM, causing cartilage damage and changing its biomechanical characteristics. MMPs are a type of protease involved in bone formation, angiogenesis, and connective tissue remodeling. In osteoarthritic cartilage, MMP-2, MMP-9, and MMP-13 have been found to be considerably overexpressed [8]. MMP-2 is a proteinase that degrades undamaged type IV collagen and type I collagen that has been denatured in the extracellular compartment. MMP-2 mediated tissue remodeling plays a role in several physiological mechanisms, such as angiogenesis, neovascularization, and wound healing. Underactivity of MMP-2, either deficiency or insufficiency, has been linked to inflammation, metabolic dysregulation, and skeletal diseases [9,10].

MMP-8 can cleave type I–III collagens. MMP-8 has been found to participate in the breakdown of ECM and the degeneration of bone tissue. MMP-8 has the ability to generate and remodulate tissue via ECM breakdown, implying that MMP-8 could be an indication of active cartilage and might have a protective function [8]. MMP-8 and MMP-9 overexpression increased cartilage damage and promoted degenerative alterations in knee structure and morphology in particular. In diabetic osteoarthritis (OA) rats, the overexpression of MMP8 and MMP9 enhanced the number of apoptotic chondrocytes [11].

MMP-9 at increased levels, adds to cartilage degradation. Pro-inflammatory indicators including interleukin-1 (IL-1), interleukin-6 (IL-6), and C-reactive protein increase its expression [12].

TIMPs have the capacity to stop MMPs from functioning. Furthermore, irrespective of their MMP neutralizing actions, TIMPs are thought to have implications of pluripotency on cellular functions, such as cell proliferation, movement, endurance, and differentiation [13]. Tissue inhibitor of metalloproteinases 2 (TIMP-2) expression is lower in studies with pregnant pig mothers receiving hydroxy—methobutyrate supplementation for the skeletal development of their piglets, which could imply that these supporting tissues of animals are undergoing greater remodeling [14]. The effect of a combination of anabolic growth factors and a protease inhibitor on an in vitro culture of articular chondrocytes revealed that TIMP-2 was beneficial in enhancing ECM synthesis [15].

IL-1 is one of the most important pro-inflammatory cytokines implicated in cartilage degradation. It draws monocytes and neutrophils to the site of tissue damage, inducing MMPs, and disrupting homeostasis. IL-1 stimulates the production of other cytokines, such as IL-1, IL-6, and TNF- during the inflammatory process [16]. This cytokine can trigger a slew of catabolic mediators in chondrocytes, the majority of which are involved in cartilage destruction, decreasing matrix formation, and chondrocyte apoptosis after a traumatic damage [17,18].

Chondrocytes can synthesize interleukin-10 (IL-10) and have the IL-10 receptor expressed on their cellular surface. In mechanically wounded cartilage, IL-10 therapy reduces posttraumatic cell death, matrix degradation, and chondrocyte dedifferentiation. In experimental models of OA, IL-10 inhibited matrix degrading enzymes and IL-1b expression produced by proinflammatory cytokines such as tumor necrosis factor alpha (TNF-$\alpha$). In addition, IL-10 enhanced proteoglycan production in an inflammatory setting increased the previously decreased biosynthetic activity of articular chondrocytes, displaying anti-apoptotic properties as well [19].

Basic fibroblast growth factor (bFGF) can enhance the proliferation of chondrocytes and mesenchymal cells, as well as chondrogenic differentiation of bone marrow mesenchymal stem cells (BMSCs) in the culture of chondrocytes and BMSCs for cartilage tissue engineering. bFGF increases chondrogenesis while inhibiting osteogenesis and protects cartilage from injury [20]. It has been widely used in tissue engineering to increase chondrocyte proliferation, angiogenesis and healing of wounds via influencing epithelial cells, smooth muscle cells, fibroblasts and endothelial cells as bFGF can stimulate cell mitosis, and thus cell proliferation [21].

Transforming growth factor β (TGF-β) promotes collagen, fibronectin, and proteoglycan synthesis while inhibiting collagen breakdown by lowering MMP activity and boosting TIMP activity [22]. When chondrocytes were stimulated mechanically, they produced more pro-osteoclastic factors, such as transforming growth factor 1 (TGF-1), which increased condylar subchondral bone resorption by boosting osteoclastogenesis. These findings back up the theory that cartilage alterations occur before subchondral bone modificas, and so play a key role in mechanical loading [23].

The objective of our current research is triggered by the lack of research in the cartilage tissue of CLP patients and aims to estimate the relative number and presence of tissue factors (MMP-2, MMP-8, MMP-9, TIMP-2, IL-1$\alpha$, IL-10, bFGF, and TGFβ1) in the CLP patients' cartilage throughout the first and second plastic rhinoplasty and alveolar osteoplasty.

## 2. Materials and Methods

### 2.1. Patients

When performing rhinoplasty and alveolar osteoplasty for the first and second procedures, tissue samples were taken from the nasal septum. The surgeries were performed in Cleft Lip and Palate Centre of Institute of Stomatology of the Riga Stradiņš University (RSU). The inclusion requirements for the patient group were diagnosis and following treatment of unilateral non-syndromic cleft lip, bilateral cleft lip, or isolated cleft palate, as well as no additional pathology that would prevent the patient from getting a surgery for cleft lip and palate. The control group's inclusion criteria were the following: no signs of inflammation or other pathological changes were found in the tissue sample, and neither anamnesis nor family history revealed any cases of craniofacial clefts. The first-time surgery patients group consisted of 35 CLP patients between the ages of 5 years and 7 months to 18 years and 7 months with cartilage tissue samples from the nasal septum. A total of 23 male patients and 12 female patients were included. The average age was 16 years and eight months (see Table 1). The second-time surgery patients group consisted of seven CLP patients between the ages of 7 years to 11 years and 7 months with cartilage tissue samples from nasal septum. Four male patients and three female patients were included. The average age was 7 years and 6 months (see Table 2). Children without CLP who underwent unrelated operations could not be used in the control group due to ethical reasons, therefore the material of cartilage tissue form trachea of 11 individuals between 20 and 40 years of age was obtained from the exposition of RSU Institute of Anatomy and Anthropology (AAI). The local Riga Stradiņš University Ethical Committee gave the approval to this study (Nr. 5/28 June 2018). A written consent form from the parents was obtained in each case.

**Table 1.** Information about the first-time surgery CLP patients.

| Patient | Sex | Age |
|---|---|---|
| No. 1. | Male | 17 years |
| No. 2. | Male | 13 years 1 month |
| No. 3. | Female | 18 years |
| No. 4. | Female | 12 years 11 months |
| No. 5. | Male | 13 years 11 months |
| No. 6. | Male | 13 years 11 months |
| No. 7. | Female | 16 years 10 months |
| No. 8. | Male | 16 years 7 months |
| No. 9. | Male | 16 years 7 months |
| No. 10. | Female | 5 years 7 months |
| No. 11. | Male | 16 years 3 months |
| No. 12. | Female | 12 years 7 months |
| No. 13. | Male | 13 years 9 months |
| No. 14. | Male | 13 years 9 months |
| No. 15. | Female | 12 years 11 months |

**Table 1.** *Cont.*

| Patient | Sex | Age |
|---|---|---|
| No. 16. | Male | 16 years 6 months |
| No. 17. | Female | 9 years 6 months |
| No. 18. | Female | 8 years 3 months |
| No. 19. | Male | 12 years 2 months |
| No. 20. | Male | 18 years 7 months |
| No. 21. | Female | 12 years 8 months |
| No. 22. | Female | 12 years 8 months |
| No. 23. | Male | 13 years 9 months |
| No. 24. | Female | 12 years 8 months |
| No. 25. | Male | 6 years 7 months |
| No. 26. | Male | 15 years |
| No. 27. | Female | 5 years 7 months |
| No. 28. | Male | 16 years 3 months |
| No. 29. | Male | 15 years 1 month |
| No. 30. | Male | 15 years 1 month |
| No. 31. | Male | 15 years 10 months |
| No. 32. | Male | 7 years 7 months |
| No. 33. | Male | 7 years 5 months |
| No. 34. | Male | 17 years |
| No. 35. | Male | 15 years 1 month |

**Table 2.** Information about the second-time surgery CLP patients.

| Patient | Sex | Age |
|---|---|---|
| No. 1. | Male | 8 years 3 months |
| No. 2. | Male | 8 years |
| No. 3. | Male | 11 years 6 months |
| No. 4. | Female | 11 years 9 months |
| No. 5. | Female | 11 years 9 months |
| No. 6. | Male | 7 years |
| No. 7. | Female | 9 years 4 months |

*2.2. Methods*

The Stefanini solution was used to fix the patient group's material in transport test tubes. Tyrode's solution was used for a rinse of the material for 24 h, and alcohol solution was used for the increase in tissue dewatering. For degreasing, tissue was submerged in xylene for 30 min. The tissue was then submerged in paraffin for one hour, followed by another two hours to allow for hardening. Using a semi-automatic rotary microtome, tissue blocks were divided into slices measuring 3 μm (Leica RM2245, Leica Biosystems Richmond Inc., Richmond, IL, USA). The sections were mounted on slides, dried in a thermostat, dewaxed again in xylene, dehydrated in alcoholic solutions, and stained with hematoxylin and eosin (H&E). Afterwards, for the expression of proteins, the immunohistochemistry method was used. The following antibodies were processed and stained on the selected tissue samples: matrix metalloproteinase-2 (MMP-2) (code: ORB101049, rabbit, 1:400, Biorbyt, Miramar Beach, FL, USA), matrix metalloproteinase-8 (MMP-8) (code: orb18114, rabbit, 1:100, Biorbyt, USA), matrix metalloproteinase-9 (MMP-9) (code: orb11064, rabbit, 1:100, Biorbyt, USA), tissue inhibitor of metalloproteinases-2 (TIMP-2) (code: SC-21735, mouse, 1:50, Santa Cruz, CA, USA), interleukin-1 alpha (IL-1α) (code: orb308787, mouse, 1:100, Biorbyt, MI, USA), interleukin-10 (IL-10) (code: orb100193, rabbit, 1:600, Biorbyt, USA), basic fibroblast growth factor (bFGF) (code: AB16828, rabbit, 1:200, Abcam, Cambridge UK) and transforming growth factor beta 1 (TGFβ1) (code: ORB7087, rabbit, 1:200, Biorbyt, USA). The semi-quantitative census technique was implemented to quantify immunological structures [24]. The labels were as follows: (0)—there was no visible positive structure in the visual array, (0/+)—visual array occasionally showing positive chondrocytes, (+)—few

positive chondrocytes are visible in the visual array, (+/++)—few to moderate number of positive chondrocytes are visible in the visual array, (++)—moderate number of positive chondrocytes seen are visible in the visual array, (++/+++)—moderate to numerous positive chondrocytes are visible in the visual array, (+++)—numerous positive chondrocytes are visible in the visual array, (+++/++++)—numerous to an abundance of positive chondrocytes are visible in the visual array, (++++)—an abundance of positive chondrocytes are visible in the visual array.

### 2.3. Statistical Analysis of the Data

The SPSS 22.0 program version (IBM Corp., Armonk, NY, USA) was used for the statistical analysis. The correlations between values were calculated using Spearman's rank correlation coefficient ($r_s$). The findings were interpreted as: $r_s$ = 0.4–0.59—moderate, positive correlation, $r_s$ = 0.6–0.79—strong, positive correlation. The study groups were compared using the Mann-Whitney U-test, but a normality test was not performed; a $p$ value < 0.05 was considered statistically significant.

### 3. Results

### 3.1. MMP-2

MMP-2 positive cells were detected in all cartilage tissue samples from the first-time surgery CLP group, the second-time surgery CLP group and the control group. The first-time surgery CLP group had a range of MMP-2 positive chondrocyte numbers, from + to +++, in the second-time surgery CLP group it varied from ++/+++ to +++/++++, and from +/++ to +++ in the control group (see Table 3) (see Figure 1A,A1).

**Table 3.** The most common relative number of factor positive chondrocytes in the first-time surgery CLP patients, the second-time surgery CLP patients and the control group.

| | MMP-2 | MMP-8 | MMP-9 | TIMP-2 | IL-1α | Il-10 | bFGF | TGFB1 |
|---|---|---|---|---|---|---|---|---|
| **First-time surgery CLP** | +/++–++++ | ++–++++ | ++–++++ | +/++–++++ | ++–++++ | ++–++++ | +–++++ | 0/+–++++ |
| Median value | +++ | +++/++++ | +++ | ++ | +++/++++ | ++/+++ | +++ | +++ |
| **Second-time surgery CLP** | ++/+++– +++/++++ | ++– +++/++++ | ++/+++– +++/++++ | ++/+++– +++/++++ | ++–++++ | ++/+++– +++/++++ | +++– +++/++++ | ++–++++ |
| Median value | +++ | +++ | +++ | +++ | +++ | +++ | +++/++++ | +++/++++ |
| **Control group** | +/++–+++ | ++/+++– ++++ | +–+++ | +/++– +++/++++ | ++–++++ | +/++–+++ | +/++–+++ | +/++– +++/++++ |
| Median value | ++ | +++/++++ | +/++ | ++/+++ | ++/+++ | ++ | ++ | ++/+++ |

Abbreviations: MMP-2—Matrix metalloproteinase-2; MMP-8—Matrix metalloproteinase-8; MMP-9—Matrix metalloproteinase-9; TIMP-2—tissue inhibitor of metalloproteinases-2; IL-1α—Interleukin-1 alpha; IL-10—Interleukin-10; bFGF—basic fibroblast growth factor; TGFB1—transforming growth factor beta 1. Median value—middle number in a sorted list of numbers. Quantification of Structures: (0)—there was no visible positive structure in the visual array (0/+)—visual array occasionally showing positive chondrocytes (+)—few positive chondrocytes are visible in the visual array, (+/++)—few to moderate number of positive chondrocytes are visible in the visual array, (++)—moderate number of positive chondrocytes seen are visible in the visual array, (++/+++)—moderate to numerous positive chondrocytes are visible in the visual array, (+++)—numerous positive chondrocytes are visible in the visual array, (+++/++++)—numerous to an abundance of positive chondrocytes are visible in the visual array, (++++)—an abundance of positive chondrocytes are visible in the visual array.

The medial value of the positive cells for MMP-2 in the first-time surgery and the second-time surgery CLP group was +++ (SD = 0.89; SD = 0.41); however, in the control group it was lower—++ (SD = 0.46) number of positive structures (see Table 3).

A statistically significant greater number of the positive chondrocytes for MMP-2 was found in the first-time surgery CLP group relative to the control group (U = 92.0; $p$ = 0.011) and in the second-time surgery CLP group in comparison to the control group (U = 7.5; $p$ = 0.003). The first-time surgery CLP group and the second-time surgery CLP group did not statistically significantly differ from one another (U = 117.0; $p$ = 0.959).

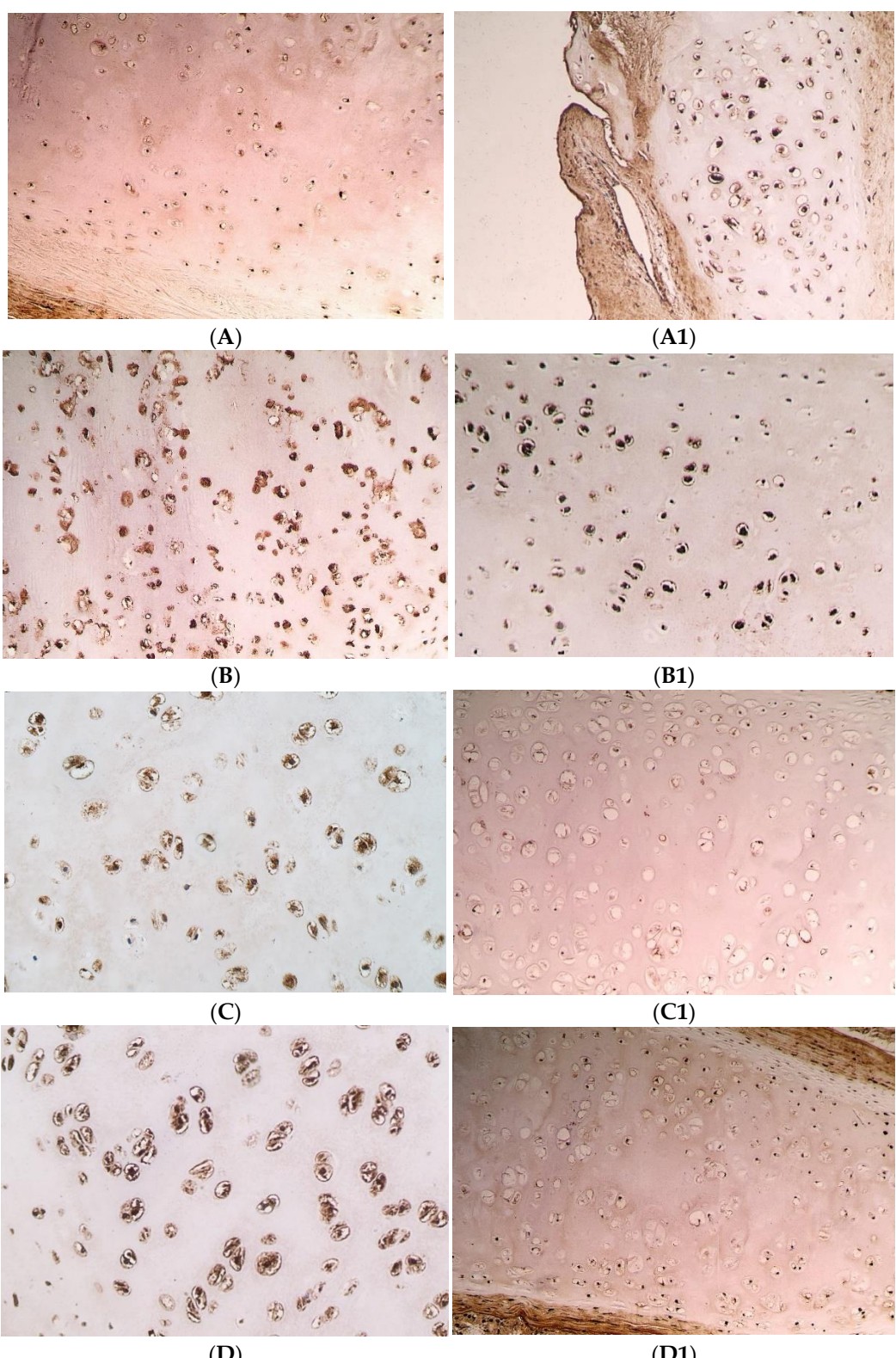

**Figure 1.** Immunohistochemical microphotographs of relative number of cells of various factors in cartilage in the CLP patients and the control patients. (**A**) +++ MMP-2 positive chondrocytes found in the cartilage of 8 years and 3 months old CLP patient from second-time surgery, IMH, ×200. (**A1**) ++ MMP-2 positive chondrocytes found in the cartilage of a control patient, IMH, ×100. (**B**) +++ MMP-8 positive chondrocytes found in the cartilage of 7 years and 2 months old CLP patient from second-time

surgery, IMH, ×200. (**B1**) +++/++++ MMP-8 positive chondrocytes found in the cartilage of a control patient, IMH, ×200. (**C**) +++ MMP-9 positive chondrocytes found in the cartilage of 11 years and 6 months old CLP patient from second-time surgery, IMH, ×250. (**C1**) +/++ MMP-9 positive chondrocytes found in the cartilage of a control patient, IMH, ×200. (**D**) +++ TIMP-2 positive chondrocytes found in the cartilage of 11 years and 6 months old CLP patient from second-time surgery, IMH, ×250. (**D1**) ++/+++ TIMP-2 positive chondrocytes found in the cartilage of a control patient, IMH, ×200.

### 3.2. MMP-8

MMP-8 positive cells were detected in all cartilage tissue samples from the first-time surgery CLP group, the second-time surgery CLP group, and the control group. The number of MMP-8 positive chondrocytes in the first-time surgery CLP group differed from ++ to ++++, in the second-time surgery CLP group—from ++/+++ to ++++, and in the control group—from ++ to +++/++++ (see Table 3) (see Figure 1B,B1).

The medial value of the MMP-8 positive cells in the first-time surgery CLP group and the control group was +++/++++ (SD = 0.68; SD = 0.45), but in the second-time surgery CLP group it was lower—+++ (SD = 0.53) number of positive structures (see Table 3).

No statistically significant difference between all of the research groups was obtained. Between the first-time surgery and the second-time surgery CLP groups (U = 81.5; $p$ = 0.198), between the first-time surgery CLP group and the control group (U = 168.5; $p$ = 0.630), and between the second-time surgery CLP group and the control group (U = 19.5; $p$ = 0.085).

### 3.3. MMP-9

All specimens had cells that were MMP-9 positive. The number of MMP-9 positive chondrocytes in the first-time surgery CLP group were +++/++++, in the second-time surgery CLP group it was ++/+++ to +++/++++, and it ranged from + to +++ in the control group (see Table 3) (see Figure 1C,C1).

The medial value of MMP-9 positive chondrocytes was the same in the first-time surgery and the second-time surgery CLP groups—+++ (SD = 0.58; SD = 0.38), but in the control group it was lower—+/++ (SD = 0.52) (see Table 3).

The first-time surgery CLP group and the second-time surgery CLP group did not have any statistically significant difference between the numbers of MMP-9 positive chondrocytes (U = 117.0; $p$ = 0.959). Both the first-time surgery CLP group and the second-time surgery CLP group had significantly more MMP-9 positive chondrocytes than the control group, (U = 31.5; $p$ = 0.000) and (U = 4.5; $p$ = 0.001), respectively.

### 3.4. TIMP-2

Each sample of cartilage tissue revealed cells that were TIMP-2 positive. Chondrocytes that were TIMP-2 positive in the first-time surgery CLP group differed from +/++ to ++++, in the second-time surgery CLP group—from ++/+++ to +++/++++, while in the control group it differed from +/++ to +++/++++ (see Table 3) (see Figure 1D,D1).

The medial value of the first-time surgery CLP group was lower than in the other groups—a ++ (SD = 0.65) number of TIMP-2 positive chondrocytes compared to +++ (SD = 0.29) in the second-time surgery CLP group, and ++/+++ (SD = 0.55) in the control group (see Table 3).

Comparing the second-time surgery CLP group to the control group, it was found that there were considerably more TIMP-2 positive chondrocytes in the second-time surgery CLP group (U = 16.0; $p$ = 0.044). When comparing the first-time surgery CLP group to the second-time surgery CLP group—a substantial increase in the number of TIMP-2 positive chondrocytes was seen (U = 53.5; $p$ = 0.021). Between the first-time surgery CLP group and the control group, there was no statistically significant difference (U = 162.5; $p$ = 0.523).

### 3.5. IL-1α

IL-1α positive chondrocytes were found in every cartilage tissue sample. The number of IL-1α positive chondrocytes in all the groups varied from ++ to ++++ (see Table 3).

The medial value of the IL-1α positive cells differed in all groups—+++/++++ (SD = 0.68) in the first-time surgery CLP group, +++ (SD = 0.63) in the second-time surgery CLP group and ++/+++ (SD = 0.19) in the control group (see Table 3) (see Figure 2A,A1).

No differences were found between the first-time surgery CLP group and the second-time surgery CLP group (U = 97.5; *p* = 0.407), the first-time surgery CLP group and the control group (U = 165.5; *p* = 0.476), and between the second-time surgery CLP group and the control group (U = 23.5; *p* = 0.179).

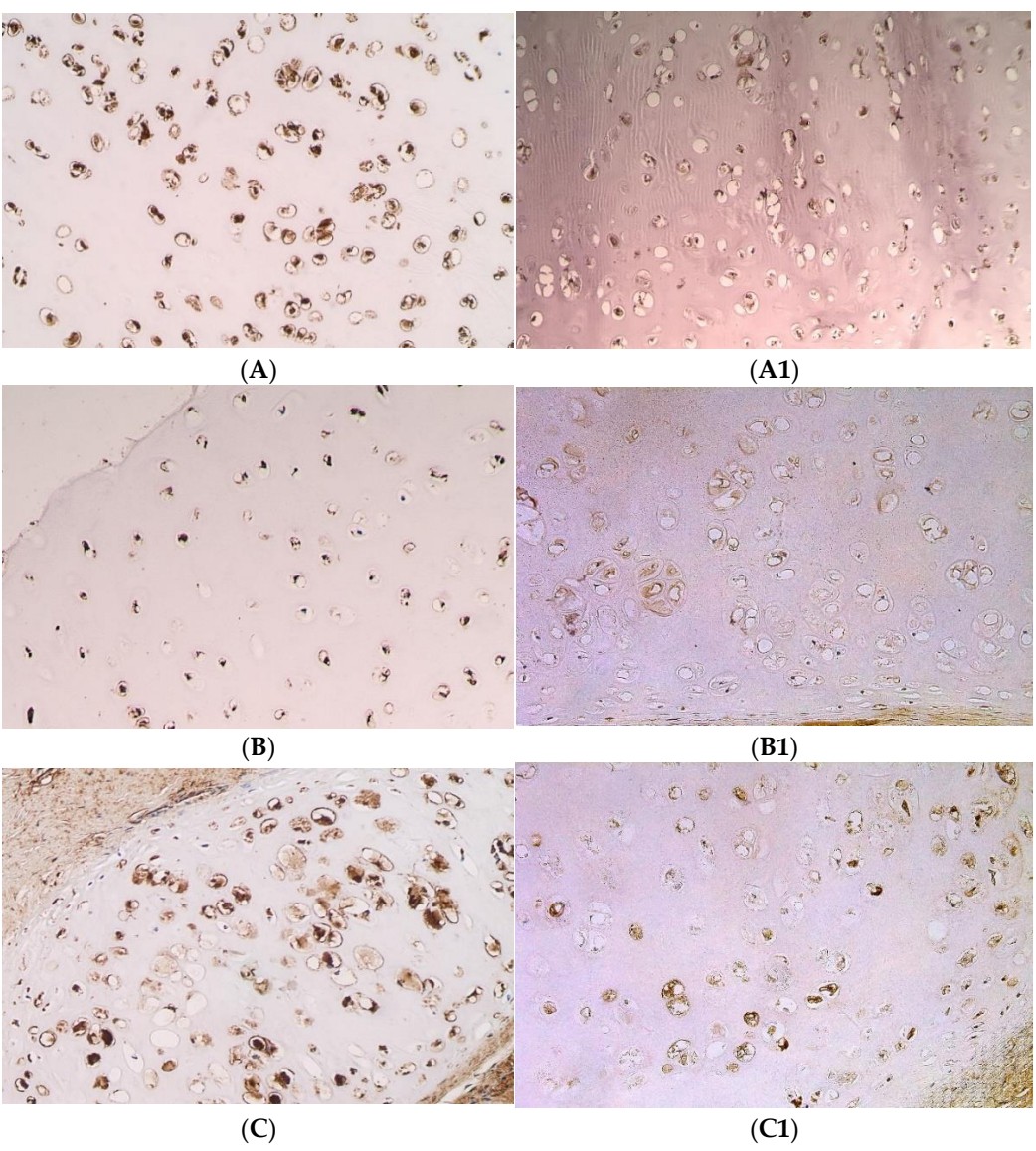

**(A)** **(A1)**

**(B)** **(B1)**

**(C)** **(C1)**

**Figure 2.** *Cont.*

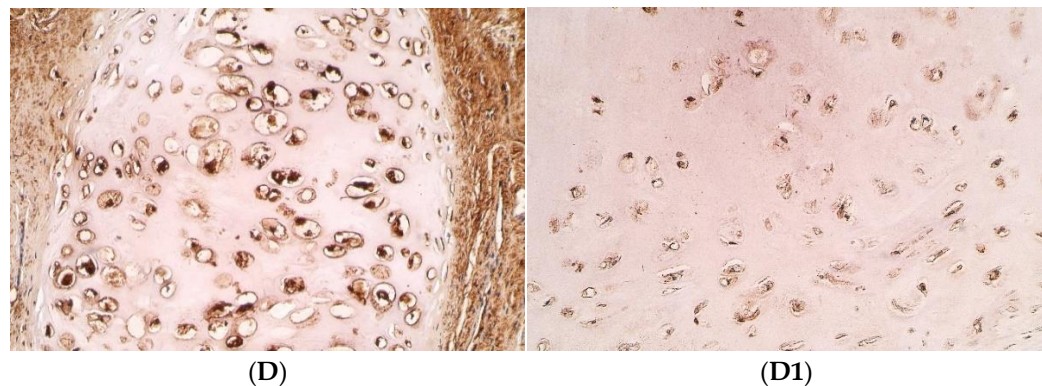

(**D**)                                    (**D1**)

**Figure 2.** Immunohistochemical microphotographs of relative number of cells of various factors in cartilage in the CLP patients and the control patients. (**A**) +++ IL-1α positive chondrocytes found in the cartilage of 11 years and 9 months old CLP patient from second-time surgery, IMH, ×250. (**A1**) ++ IL-1α positive chondrocytes found in the cartilage of a control patient, IMH, ×250. (**B**) +++ IL-10 positive chondrocytes found in the cartilage of 8 years old CLP patient from second-time surgery, IMH, ×200. (**B1**) ++ IL-10 positive chondrocytes found in the cartilage of a control patient, IMH, ×200. (**C**) +++/++++ bFGF positive chondrocytes found in the cartilage of 8 years and 3 months old CLP patient from second-time surgery, IMH, ×250. (**C1**) ++ bFGF positive chondrocytes found in the cartilage of a control patient, IMH, ×250. (**D**) +++ TGFβ1 positive chondrocytes found in the cartilage of 8 years and 3 months old CLP patient from second-time surgery, IMH, ×250. (**D1**) ++/+++ TGFβ1 positive chondrocytes found in the cartilage of a control patient, IMH, ×250.

### 3.6. IL-10

Cells that were IL-10 positive were found in all cartilage samples, and the number of IL-10 positive chondrocytes in the first-time surgery CLP group was ++ to ++++, in the second-time surgery CLP group it was ++/+++ to +++/++++, and in the control group—from +/++ to +++ (see Table 3) (see Figure 2B,B1).

The medial value of IL-10 positive cells was ++/+++ (SD = 0.62) in the first-time surgery CLP group and +++ (SD = 0.35) in the second-time surgery CLP group, in comparison to ++ (SD = 0.16) in the control group (see Table 3).

Between the first-time surgery CLP group and the control group, a statistically significant difference was discovered (U = 82.0; $p$ = 0.004), as well as between the second-time surgery CLP group and the control group (U = 7.0; $p$ = 0.003), where the count of IL-10 positive cells was higher in the CLP groups in comparison to the control group. However, no statistically significant difference between the first-time surgery and the second-time surgery CLP groups was found (U = 87.5; $p$ = 0.243).

### 3.7. bFGF

All cartilage tissue samples from the CLP groups and the control group had bFGF positive cells. The number of bFGF positive chondrocytes in the first-time surgery CLP group was the most variable—from + to ++++, in the second-time surgery CLP group it varied from +++ to +++/++++, while in the control group it ranged from +/++ to +++ (see Table 3) (see Figure 2C,C1).

The medial value of the first-time surgery CLP group's positive chondrocytes was +++ (SD = 0.87), in the second-time surgery CLP group—+++/++++ (SD = 0.27), compared to ++ (SD = 0.60) in the control group (see Table 3).

The first-time surgery CLP group (U = 80.5; $p$ = 0.005) and the second-time surgery CLP group (U = 4.500; $p$ = 0.001) had a considerably larger number of chondrocytes that were bFGF positive when compared to the control group. The number of chondrocytes that were positive for bFGF in the first-time surgery CLP group and the second-time surgery CLP group did not significantly differ statistically (U =1 01.5; $p$ = 0.626).

*3.8. TGFβ1*

All cartilage tissue samples from the CLP groups and the control group contained TGFβ1 positive cells. In the CLP groups, the quantity of TGFβ-1 positive chondrocytes ranged from 0/+ to +++, whereas in the control group, it ranged from +/++ to +++/++++ (see Table 3) (see Figure 2D,D1).

The medial value of TGFβ1 positive chondrocytes in the first-time surgery CLP group was +++ (SD = 0.88), while in the second-time surgery CLP group it was +++/++++ (SD = 0.64), and in the control group there was a ++/+++ (SD = 0.66) number of TGFβ1 positive chondrocytes (see Table 3).

A higher amount of TGFβ1 positive chondrocytes was observed in the second-time surgery CLP group compared to the control group (U = 14.0; $p = 0.027$). There was no difference observed between the first-time surgery CLP group and the control group (U = 163.5; $p = 0.461$) and between the first-time surgery CLP group and the second-time surgery CLP group (U = 69.0; $p = 0.073$).

*3.9. Statistical Data*

Statistically significant ($p < 0.05$) strong ($r_s = 0.6$–$0.79$) correlations were found between IL-1α and IL-10 ($r_s = 0.698$; $p = 0.000$); IL-1α and MMP-8 ($r_s = 0.604$; $p = 0.000$) in the first-time surgery CLP group (see Table 4).

**Table 4.** An overview of the Spearman's rank correlation analysis, conducted to identify the moderate and strong links between the number of positive factors in the first-time surgery CLP group.

| Factor 1 | Factor 2 | $p$ Value | $r_s$ |
|---|---|---|---|
| **Statistically Significant ($p < 0.05$) Strong Correlations ($r_s = 0.6$–$0.79$)** | | | |
| **IL-1α** | IL-10 | $p = 0.000$ | $r_s = 0.698$ |
| **IL-1α** | MMP-8 | $p = 0.000$ | $r_s = 0.604$ |

Abbreviations: $r_s$—Spearman's rank correlation coefficient; IL-1α—Interleukin-1 alpha; IL-10—Interleukin-10; MMP8—Matrix metalloproteinase-8.

Statistically significant ($p < 0.05$) strong ($r_s = 0.6$–$0.79$) correlation was found between IL-1α and IL-10 ($r_s = 0.877$; $p = 0.010$) in the second-time surgery CLP group (see Table 5).

**Table 5.** An overview of the Spearman's rank correlation analysis, conducted to identify the moderate and strong links between the number of positive factors in the second-time surgery CLP group.

| Factor 1 | Factor 2 | $p$ Value | $r_s$ |
|---|---|---|---|
| **Statistically Significant ($p < 0.05$) Strong Correlations ($r_s = 0.6$–$0.79$)** | | | |
| **IL-1α** | IL-10 | $p = 0.010$ | $r_s = 0.877$ |

Abbreviations: $r_s$—Spearman's rank correlation coefficient; IL-1α—Interleukin-1 alpha; IL-10—Interleukin-10.

Statistically significant ($p < 0.05$) strong ($r_s = 0.6$–$0.79$) correlations were found between MMP-2 and TIMP-2 ($r_s = 0.867$; $p = 0.001$); MMP-2 and MMP-8 ($r_s = 0.743$; $p = 0.009$); MMP-9 and IL-1α ($r_s = 0.850$; $p = 0.001$) in the control group (see Table 6).

**Table 6.** An overview of the Spearman's rank correlation analysis, conducted to identify the moderate and strong links between the number of positive factors in the control group.

| Factor 1 | Factor 2 | $p$ Value | $r_s$ |
|---|---|---|---|
| **Statistically Significant ($p < 0.05$) Strong Correlations ($r_s = 0.6$–$0.79$)** | | | |
| **MMP-2** | TIMP-2 | $p = 0.001$ | $r_s = 0.867$ |
| **MMP-2** | MMP-8 | $p = 0.009$ | $r_s = 0.743$ |
| **MMP-9** | IL-1α | $p = 0.001$ | $r_s = 0.850$ |

Abbreviations: $r_s$—Spearman's rank correlation coefficient; IL-1α—Interleukin-1 alpha; MMP-2—Matrix metalloproteinase-2 MMP-8—Matrix metalloproteinase-8; MMP-9—Matrix metalloproteinase-9.

## 4. Discussion

Chondrocytes are highly specific, differentiated cells whose primary role is to preserve the stability of cartilage matrix elements. The structure and organization of cartilage's ECM are the essential components of appropriate function. It can be destroyed by illness or trauma, which limits the ability of the body to self-repair. On account of cell-based research and treatment, the demand for cartilage cells is quickly increasing. However, there is still a lack of full understanding of pathophysiology and delays in diagnosis due to the avascular character of cartilage, the neural structure of the tissue, and obstacles connected with providing potential therapies [25,26].

In this research, we aimed to discover more about the mechanisms behind cartilage remodeling in CLP patients, as well as the potential outcomes of the tissue healing after surgical intervention, and how these mechanisms are connected to the tissue factors we have looked at. The current data were obtained from a mixed group of CLP patients with unilateral cleft lip and palate and bilateral cleft lip and palate.

### 4.1. Tissue Degradation Factors and Their Inhibitors

In the present research, tissue degradation factors MMP-2, MMP-8, and MMP-9 and their inhibitor TIMP-2 were investigated. Our study results demonstrated a statistically significant ($p < 0.05$) increase in MMP-2 and MMP-9 in CLP patients, both first-time surgery and second-time surgery CLP groups, when compared to the control group. This would suggest that CLP patients have more pronounced nasal cartilage remodeling and repair. This could imply that tissue is more requisitioned for tissue repair, and greater amounts of MMP-2 positive cells could signal the start of the healing process following surgery. MMPs-2 and -9 are thought to cause extensive extracellular degradation in vivo via epithelial mesenchymal interactions [27]. MMP-2 deficiency can result in skeletal development problems, such as lack of bone mineral density, cartilage destruction, and aberrant craniofacial development [28]. There is a lack of research in the field about the hyaline cartilage of the nasal septum. However, in the articular cartilage, MMP-2 and MMP-9 have been studied in osteoarthritis (OA) and normal cartilage samples. When compared to normal articular cartilage specimens, all the OA specimens had a higher frequency and distribution of each MMP [29]. Additionally, a low level of one MMP can be countered by increasing the level of another MMP. For example, in a wound repair model lacking MMP-8, the MMP-9 expression was observed to increase [30]. Although there were no statistically significant ($p < 0.05$) differences in MMP-8 levels between the CLP groups and the control group, there were statistically significant strong positive correlations between IL-1 and MMP-8 ($r_s = 0.604$; $p = 0.000$) for the first-time surgery CLP patients.

TIMP-2 was investigated, and statistically significant ($p < 0.05$) differences between the study groups were observed between the first-time surgery CLP group and the second-time surgery CLP group, where the first-time surgery CLP group presented a higher amount of TIMP-2 than the second-time surgery CLP group (U = 53.5; $p = 0.021$). It was discovered that statistically significant TIMP-2 elevation in primary operated cartilage indicates the selective tissue remodulation impact for hard tissue. MMPs activity is strongly regulated by the TIMPs. In many cases, matrix turnover is impacted indirectly by the MMP/TIMP axis' control of these physiologically active proteins. TIMPs have been identified to regulate a variety of processes, including modified transforming growth factor (TGF) signaling, inflammation, or the quantity of myofibroblast-like cells, all of which have the ability to alter ECM turnover and promote ECM deposition. These results imply that TIMPs can indirectly regulate ECM turnover while also directly inhibiting ECM proteolysis. Numerous severe disorders including cancer and rheumatoid arthritis are linked to an imbalance in this system that results in elevated MMP activity [31]. MMPs have important functions in embryogenesis and the homeostasis of adult tissues along with their TIMPs, which are associated with the CLP [32]. Due to the suppression of the degradative processes, high TIMP concentrations cause ECM to accumulate, whereas low TIMP activity causes enhanced proteolysis. By forming stoichiometric complexes, all four TIMP family members

(TIMP-1, -2, -3, -4) block the corresponding enzyme [33]. Additionally, vascular endothelial growth factor and FGF-2 responses are inhibited by TIMP-2 in a manner that is MMP independent. The tissue of the CLP demonstrates altered correlations in the amount, favoring the TIMPs level [34].

*4.2. Interleukins*

In this study, the interleukins IL-1 and IL-10 were examined. It was observed that the numbers of IL-1$\alpha$ positive osteocytes were +++/++++ in the first-time surgery CLP group and +++ in the second-time surgery CLP group. Additionally, there was no statistically significant difference ($p < 0.05$) between the CLP groups and the control group, indicating that pro-inflammatory cytokines such IL-1 may not be a main factor in the pathogenesis of CLP. Members of the IL-1 family of cytokines are fundamental mediators of inflammation. IL-1 is a strong pro-inflammatory cytokine that draws neutrophils and monocytes to the site of tissue injury and activates MMPs. IL-1 stimulates the production of other cytokines such as IL-1, IL-6, and TNF- during the inflammatory process [16]. It is a bioactive mediator that is constitutively present in almost all cell types and is released following cell death or expressed by myeloid cells entering wounded tissues. Since IL-1 is a bioactive precursor that is generated after cell death and tissue injury, it plays a crucial role in the etiology of many diseases that are characterized by organ or tissue inflammation [35]. IL-1 enhances the breakdown of cartilage, indicating that chondrocytes are stimulated by IL-1 to break down their own matrix [36]. IL-1 levels associated with OA reprogram articular chondrocytes, making them more vulnerable to mechanical damage, resulting in an excess of intracellular Ca2+ both at rest and in response to mechanical deformation [37].

The present study showed a statistically significant ($p < 0.05$) increased amount of IL-10 found in the first-time surgery CLP group (U = 82.0; $p = 0.004$) and the second-time surgery CLP group (U = 7.0; $p = 0.003$) in comparison to the control group. It was found that IL-10 positive cells made a strong, statistically significant positive correlation with IL-1$\alpha$, in both the first-time surgery CLP group ($r_s = 0.698$; $p = 0.000$) and the second-time surgery CLP group ($r_s = 0.877$; $p = 0.010$), most likely indicating the compensatory anti-inflammatory protective mechanism in the cartilage of patients with CLP. IL-10, in contrast to IL-1$\alpha$, is an anti-inflammatory cytokine that can inhibit the release of inflammatory factors [38]. Treatment with IL-10 inhibits the synthesis of IL-1, IL-6, and IL-8. IL-10 efficiently enhanced type II collagen and Sox9 mRNA expression, as well as aggrecan and type II collagen protein expression [39]. In autologous chondrocyte transplantation, overexpressed IL-10 maintained a more healing, long-lasting, and obvious effect in post-traumatic cartilage [40].

*4.3. Growth Factors*

In this research, the growth factors TGF$\beta$1 and bFGF were investigated. We detected a statistically significant ($p < 0.05$) rise in bFGF quantities in the first-time surgery CLP patients (U = 80.5; $p = 0.005$) and the second-time surgery CLP patients (U = 4.500; $p = 0.001$) in comparison to the control group. Additionally, it was found that the increased proportions of chondrocytes in CLP groups that express bFGF may indicate a better potential for efficient wound healing. FGF signaling pathways are crucial regulators of vertebrate skeletal development. They control chondrogenesis, osteogenesis, and mineral homeostasis, as well as wound healing, angiogenesis, tissue repair, and regeneration [41,42]. All phases of palatogenesis, particularly during palatal fusion when cells proliferate, are influenced by FGF [43]. Members of the FGF family, in particular bFGF, attach to cell surface receptors, promoting anabolic pathways, and reducing aggrecanase activity. In mice, subcutaneous administration of bFGF suppressed OA, while mice with knock-out bFGF were found to have accelerated OA [44].

Our findings showed that the second-time surgery CLP group and the control group differed in a statistically significant way (U = 14.0; $p = 0.027$), ($p < 0.05$) and the numbers of TGF$\beta$1 positive chondrocytes were +++/++++ in the second-time surgery CLP group, whereas in the control group the numbers were lower—++/+++ TGF$\beta$1 positive chondro-

cytes. Our results would suggest that the higher TGFβ1 numbers found in the cartilage of the second-time surgery CLP patients may indicate improved tissue regeneration abilities. TGF-β signaling is a critical regulator for cartilage tissue maintenance and chondrocyte homeostasis. The regulatory systems that control TGF-'s chondroprotective activity are not yet fully clarified. This cascade regulates postnatal articular cartilage homeostasis by TGF-β via a mechanism that regulates the induction of anabolic and autophagy-related gene expression in chondrocytes [45]. It has been demonstrated that the TGF-signaling pathway plays a significant role during palatogenesis [46]. TGF-β has been linked to wound healing and tissue homeostasis maintenance, by stimulating the production of collagens I and III, fibronectin and laminin, it influences cell proliferation, extracellular matrix creation, and accelerates the process of tissue remodeling [47]. When TGF levels are high, cartilage homeostasis is disturbed, and chondrocyte metabolic function is compromised [48].

## 5. Conclusions

Increased numbers of MMP-2 and MMP-9 positive chondrocytes in CLP patients could signal the beginning of the healing process following the surgery, and stimulation by the increased numbers of bFGF may also indicate a better potential for efficient wound healing.

Statistically significant TIMP-2 elevation in primary operated cartilage indicates the selective tissue remodulation impact for hard tissue, and the higher TGFβ1 numbers may indicate improved tissue regeneration abilities.

Elevated numbers and positive correlations of IL-1α and IL-10 in cleft affected tissue probably indicate the persistent inflammation process and the subsequent compensatory anti-inflammatory protective mechanism in the cartilage of patients with CLP.

**Author Contributions:** Conceptualization, D.B. and M.P.; methodology, M.P.; validation, M.P. and I.A.; formal analysis, D.B.; investigation, D.B.; resources, M.P. and I.A.; writing—original draft preparation, D.B.; writing—review and editing, D.B., M.P. and I.A.; supervision, M.P. and I.A. All authors have read and agreed to the published version of the manuscript.

**Funding:** This research received no external funding.

**Institutional Review Board Statement:** The study was conducted in accordance with the Declaration of Helsinki and approved by the Ethics Committee of Riga Stradiņš University (Nr. 5/28 June 2018).

**Informed Consent Statement:** Informed consent was obtained from all subjects involved in the study.

**Data Availability Statement:** Not applicable.

**Acknowledgments:** Rīga Stradiņš University kind support for the research is highly acknowledged.

**Conflicts of Interest:** The authors declare no conflict of interest. The funders had no role in the design of the study; in the collection, analyses, or interpretation of data; in the writing of the manuscript; or in the decision to publish the results.

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
