# Peer review of "Evaluation of the Multiple Tissue Factors in the Cartilage of Primary and Secondary Rhinoplasty in Cleft Lip and Palate Patients"

_pediatrrep, doi:10.3390/pediatric14040050_

Round 1
Reviewer 1 Report
This paper is very well written.
I have some small comments :
what do you mean with
'first time' group and 'second time' group ?
I would suggest to start with the inclusion criteria and then the composition of the groups. This explains that in the groups you took UCLP and BCLP patients.
This last point should be mentioned in the discussion that current data were found in this mixed group.
It should also be mentioned where the cartilage was taken ? Left right ?
Author Response
What do you mean with 'first time' group and 'second time' group ?
Thank you for your question! It was meant for a first-time surgery group and a second-time surgery group. For first-time group and second-time group we added the word ‘’surgery’’ throught the whole paper for better understanding!
I would suggest to start with the inclusion criteria and then the composition of the groups. This explains that in the groups you took UCLP and BCLP patients. This last point should be mentioned in the discussion that current data were found in this mixed group.
Thank you for the suggestion! We changed the order in section of ‘’Materials and methods’’ and put the inclusion criteria before the composition of the groups, and added your suggestion about the mixed groups in the discussion section.
Materials and methods
When performing rhinoplasty and alveolar osteoplasty for the first and second procedures, tissue samples were taken. Surgeries were performed in Cleft Lip and Palate Centre of Institute of Stomatology of the Riga Stradiņš University (RSU). The following were the patient groups' inclusion requirements: diagnosis and treatment of unilateral non-syndromic cleft lip, bilateral cleft lip, and isolated cleft palate, as well as no additional pathology that would prevent the patient from getting surgery for cleft lip and palate. Children with CLP who underwent unrelated operations had no chance of providing hyaline cartilage for the control group due to ethical reasons, therefore from 11 individuals between the 20 to 40 years of age the material of cartilage tissue from trachea was obtained from the exposition of RSU Institute of Anatomy and Anthropology (AAI). The control group inclusion criteria were the following: in the tissue sample, no signs of inflammation or other pathological changes were found and neither anamnesis nor family history revealed any cases of craniofacial clefts. The first-time surgery surgery patients group consisted of 35 CLP patients between the ages of 5 years and 7 months to 18 years and 7 months with cartilage tissue samples from the nasal septum. 23 male patients and 12 female patients were included. The average age was 16 years and eight months (see Table 1). The second-time surgery patients group consisted of seven CLP patients between the ages of 7 years to 11 years and 7 months with cartilage tissue samples from nasal septum. Four male patients and three female patients were included. The average age was 7 years and 6 months (see Table 2).
Discussion
In this research, we aimed to discover more about the mechanisms behind cartilage remodeling in CLP patients, as well as the potential outcomes of tissue healing after surgical intervention, and these mechanisms are connected to the tissue factors we have looked at. The current data were obtained from a mixed group of CLP patients with unilateral cleft lip and palate and bilateral cleft lip and palate.
- It should also be mentioned where the cartilage was taken? Left, right?
Thank you for your remarks! The cartilage was taken from the nasal septum; we added this information in the materials and methods section.
When performing rhinoplasty and alveolar osteoplasty for the first and second procedures, tissue samples were taken from nasal septum.

Reviewer 2 Report
The authord provide a detailed background and sufficient references.
The methodology of the study is well conducted.
Regarding the results, the mean values must be expressed in conjunction with the standard deviation.
Once this editing will be apported, I suggest the pubblication.
Author Response
Regarding the results, the mean values must be expressed in conjunction with the standard deviation.
Thank you for your suggestion! The standard deviation values were added in conjunction with the median values throughout the whole paper!

Reviewer 3 Report
Abstract should be re-written to accommodate some correlations too, not just mere presenting of data.
Please correct some minor typos in lines 96, 105, 113 (use "present research", the reader is not aware whether this is your latest research or not), line 146 ("using" instead of "applying")
For Table 3 and also throughout the whole results/discussions sections, maybe the authors can think of a numeric scale instead of +/++/+++...
Table 4 - p is not 0.000
Fig 2 - one cannot see sections G-H
The discussion section should be more clear
Author Response
Abstract should be re-written to accommodate some correlations too, not just mere presenting of data.
Thank you for the suggestion! Due to the limit of words in an abstract, we couldn’t express ourselves completely in the accommodation of data, but we changed the abstract to make it more clear and to accommodate the data better.
Abstract
CLP is one of the craniofacial defects. Our study's objective was to identify the differences and appearance between the tissue factors in cartilage of CLP patients during primary and secondary rhinoplasty. Immunohistochemistry was performed with MMP-2, MMP-8, MMP-9, TIMP-2, IL-1α, IL-10, bFGF, and TGFβ1. The quantification of structures was done using a semi-quantitative census method. Both the Mann-Whitney U test and the Spearman rank correlation coefficient were applied. MMP-2, -9, IL-1a, and bFGF demonstrated higher number of positive cells in patients, while number of MMP-8, IL-1a, -10 and TGFβ1 cells were higher or equal to the control subjects. The only statistically significant difference between CLP-operated patients was found in the TIMP-2 group, where the primary CLP patient group had a higher number of TIMP-2 positive chondrocytes than the secondary CLP patient group (U = 53.5; p = 0.021). The median value of the primary CLP group was ++ number of TIMP-2 positive chondrocytes compared to +++ in the secondary CLP group. No statistically significant difference was found between primary and secondary rhinoplasty patients for other tissue factors. Commonly, cleft affected cartilage rich expression of different tissue factors suggests the stimulation of higher elasticity of cartilage. Statistically significant TIMP-2 elevation in primary operated cartilage indicates the selective tissue remodelation impact for hard tissue.
Please correct some minor typos in lines 96, 105, 113 (use "present research", the reader is not aware whether this is your latest research or not), line 146 ("using" instead of "applying")
Thank you for your comment! The typos were corrected in line 105 and 146, however we did not find an appropriate place in the introduction in lines which you suggested (96; 113) where to add the ‘’present research’’ words.
Line 105: The objective of our present research is to estimate the relative number and presence of tissue factors (MMP-2, MMP-8, MMP-9, TIMP-2, IL-1α, IL-10, bFGF, and TGFβ1) in the CLP patients’ cartilage throughout the first and second plastic rhinoplasty and alveolar osteoplasty, based on the lack of research in the cartilage tissue of CLP patients.
Line 146: Using a semi-automatic rotary microtome, tissue blocks were divided into slices measuring 3 micrometres (Leica RM2245, Leica Biosystems Richmond Inc., United States).
For Table 3 and also throughout the whole results/discussions sections, maybe the authors can think of a numeric scale instead of +/++/+++...
Thank you for your suggestion! We used a predetermined numerical scale for the statistical evaluation, but we are using this style for simplicity, and this semi-quantitative census method with symbols +/- was also used in our previous research, so in order for the articles to be written according to a single standard, this method was applied.
Table 4 - p is not 0.000
Thank you for the remark! When we consulted the statistics department at our university, they informed us that if the p value is extremely little, statistical software will interpret it as p = 0.000, and it can be written in this way.
Fig 2 - one cannot see sections G-H
Thank you for this observation. We have corrected the formatting of the page by inserting page breaks, and now all the sections of Figure 2 are visible!
The discussion section should be more clear
Thank you for your suggestion! We have made the discussion section clearer, as we stated results before each paragraph and divided the discussion into sections.
Tissue degradation factors and their inhibitors
In the present research, tissue degradation factors MMP-2, MMP-8, and MMP-9 and their inhibitor TIMP-2 were investigated. Our study results demonstrated a statistically significant (p<0.05) increase in MMP-2 and MMP-9 in CLP patients, both first-time and second-time CLP groups, when compared to the control group. It would suggest that CLP patients have more pronounced nasal cartilage remodeling and repair. This could imply that tissue is more requisitioned for tissue repair, and greater amounts of MMP-2 positive cells could signal the start of the healing process following surgery. MMPs-2 and -9 are thought to cause extensive extracellular degradation in vivo via epithelial mesenchymal interactions [25]. MMP-2 deficiency can result in skeletal development problems such as lack of bone mineral density, cartilage destruction, and aberrant craniofacial development [26]. There is a lack of research in the field about the hyaline cartilage of the nasal septum, but in the articular cartilage, MMP-2 and MMP-9 were studied in OA and normal cartilage samples. When compared to normal articular cartilage specimens, all of the OA specimens had a higher frequency and distribution of each MMP [27]. Additionally, a low level of one MMP can be countered by increasing the level of another MMP. For example, in a wound repair model lacking MMP-8, MMP-9 expression was observed to increase [28]. Although there were no statistically significant (p<0.05) differences in MMP-8 levels between the CLP groups and the control group, there were statistically significant strong positive correlations between IL-1 and MMP-8 (rs=0.604; p=0.000) for first-time CLP patients.
TIMP-2 was investigated, and statistically significant (p<0.05) difference between the study groups were observed between the first-time CLP group and second-time CLP group, where the first-time CLP group presented a higher amount of TIMP-2 than the second-time CLP group (U=53.5; p=0.021). It was discovered that statistically significant TIMP-2 elevation in primary operated cartilage indicates the selective tissue remodelation impact for hard tissue. MMPs activity is strongly regulated by the TIMPs. In many cases, matrix turnover is impacted indirectly by the MMP/TIMP axis' control of these physiologically active proteins. TIMPs have been identified to regulate a variety of processes, including modified transforming growth factor (TGF) signalling, inflammation, or the quantity of myofibroblast-like cells, all of which have the ability to alter ECM turnover and promote ECM deposition. These results imply that TIMPs can indirectly regulate ECM turnover while also directly inhibiting ECM proteolysis. Numerous severe disorders including cancer and rheumatoid arthritis are linked to an imbalance in this system that results in elevated MMP activity [29]. MMPs have important functions in embryogenesis and the homeostasis of adult tissues along with their TIMPs, which are associated with the CLP [30]. Due to the suppression of the degradative processes, high TIMP concentrations cause ECM to accumulate, whereas low TIMP activity causes enhanced proteolysis. By forming stoichiometric complexes, all four TIMP family members (TIMP-1, -2, -3, -4) block the corresponding enzyme [31]. Additionally, vascular endothelial growth factor and FGF-2 responses are inhibited by TIMP-2. in a manner that is MMP independent. The tissue of the CLP demonstrates altered correlations in the amount in favour of the TIMPs level [32].
Interleukins
In this study, the interleukins IL-1 and IL-10 were examined. It was observed that the numbers of IL-1α positive osteocytes were +++/++++ in the first-time CLP group and +++ in the second-time CLP group. Additionally, there was no statistically significant difference (p<0.05) between the CLP groups and the control group, indicating that pro-inflammatory cytokines such IL-1 may not be a main factor in the pathogenesis of CLP. Members of the IL-1 family of cytokines are fundamental mediators of inflammation. IL-1 is a strong pro-inflammatory cytokine that draws neutrophils and monocytes to the site of tissue injury and activates MMPs. IL-1 stimulates the production of other cytokines like IL-1, IL-6, and TNF- during the inflammatory process [16]. It is a bioactive mediator that is constitutively present in almost all cell types and is released following cell death or expressed by myeloid cells entering wounded tissues. Since IL-1 is a bioactive precursor that is generated after cell death and tissue injury, it plays a crucial role in the etiology of many diseases that are characterized by organ or tissue inflammation [33]. IL-1 enhances breakdown of cartilage, that indicates that chondrocytes are stimulated by IL-1 to break down their own matrix. [34]. IL-1 levels associated with OA reprogram articular chondrocytes, making them more vulnerable to mechanical damage, resulting in an excess of intracellular Ca2+ both at rest and in response to mechanical deformation [35].
The present study showed a statistically significant (p<0.05) increased amount of IL-10 was found in the first-time CLP group (U=82.0; p=0.004) and the second-time CLP group (U=7.0; p=0.003) in comparison to the control group. It was found that IL-10 positive cells made a strong, statistically significant positive correlation with IL-1α, in both the first-time CLP group (rs=0.698; p=0.000) and the second-time CLP group (rs=0.877; p=0.010), probably indicating the compensatory anti-inflammatory protective mechanism in the cartilage of patients with CLP. IL-10, in contrast to IL-1α, is an anti-inflammatory cytokine that can inhibit the release of inflammatory factors [36]. Treatment with IL-10 inhibits the synthesis of IL-1, IL-6, and IL-8. IL-10 efficiently enhanced type II collagen and Sox9 mRNA expression, as well as aggrecan and type II collagen protein expression [37]. In autologous chondrocyte transplantation, overexpressed IL-10 maintained a more healing, long-lasting, and obvious effect in post-traumatic cartilage [38].
In our research, we detected a statistically significant (p<0.05) rise in bFGF quantities in the first-time CLP patients (U=80.5; p=0.005) and second-time CLP patients (U=4.500; p=0.001) in comparison to the control group. Additionally, it was found that the increased proportions of chondrocytes in CLP groups that express bFGF may indicate a better potential for efficient wound healing. FGF signaling pathways are crucial regulators of vertebrate skeletal development, it controls chondrogenesis, osteogenesis, and mineral homeostasis as well as wound healing, angiogenesis, tissue repair, and regeneration [39, 40]. All phases of palatogenesis, particularly during palatal fusion when cells proliferate, are influenced by FGF [41]. Members of the FGF family, in particular bFGF, attach to cell surface receptors, promoting anabolic pathways, and reducing aggrecanase activity. In mouse, subcutaneous administration of bFGF suppressed OA, while bFGF knock-out mice were found to have accelerated OA [42].
Growth factors
In this research, the growth factors TGFβ1 and bFGF were investigated. In our research, we detected a statistically significant (p<0.05) rise in bFGF quantities in the first-time surgery surgery CLP patients (U=80.5; p=0.005) and second-time surgery CLP patients (U=4.500; p=0.001) in comparison to the control group. Additionally, it was found that the increased proportions of chondrocytes in CLP groups that express bFGF may indicate a better potential for efficient wound healing. FGF signaling pathways are crucial regulators of vertebrate skeletal development, it controls chondrogenesis, osteogenesis, and mineral homeostasis as well as wound healing, angiogenesis, tissue repair, and regeneration [39, 40]. All phases of palatogenesis, particularly during palatal fusion when cells proliferate, are influenced by FGF [41]. Members of the FGF family, in particular bFGF, attach to cell surface receptors, promoting anabolic pathways, and reducing aggrecanase activity. In mouse, subcutaneous administration of bFGF suppressed OA, while bFGF knock-out mice were found to have accelerated OA [42].
Our findings showed that the second-time CLP group and the control group differed in a statistically significant way (U=14.0; p=0.027), (p<0.05) and the numbers of TGFβ1 positive chondrocytes were +++/++++ in the second-time CLP group, whereas in the control group numbers were lower - ++/+++ TGFβ1 positive chondrocytes. Our results would suggest that the higher TGFβ1 numbers found in the cartilage of second-time CLP patients may indicate improved tissue regeneration abilities. TGF-β signaling is a critical regulator for cartilage tissue maintenance and chondrocyte homeostasis. The regulatory systems that control TGF-'s chondroprotective activity are not yet fully clarified. This cascade regulates postnatal articular cartilage homeostasis by TGF- β via a mechanism that regulates the induction of anabolic and autophagy-related gene expression in chondrocytes [43]. It has been demonstrated that the TGF-signaling pathway plays a significant role during palatogenesis [44]. TGF-β has been linked to wound healing and tissue homeostasis maintenance, by stimulating the production of collagens I and III, fibronectin and laminin, it influences cell proliferation, extracellular matrix creation, and accelerates the process of tissue remodeling [45]. When TGF levels are high, cartilage homeostasis is disturbed, and chondrocyte metabolic function is compromised. [46].
Round 2
Reviewer 3 Report
The authors answered most of my concerns.
I think the manuscript requires a professional English spelling before a formal accept.
Author Response
I think the manuscript requires a professional English spelling before a formal accept.
The manuscript underwent a professional English spelling and grammar correction.
